# Hepatic Polarized Differentiation Promoted the Maturity and Liver Function of Human Embryonic Stem Cell-Derived Hepatocytes via Activating Hippo and AMPK Signaling Pathways

**DOI:** 10.3390/cells11244117

**Published:** 2022-12-18

**Authors:** Jue Wang, Ping Situ, Sen Chen, Haibin Wu, Xueyan Zhang, Shoupei Liu, Yiyu Wang, Jinghe Xie, Honglin Chen, Yuyou Duan

**Affiliations:** 1Laboratory of Stem Cells and Translational Medicine, Institutes for Life Sciences, School of Medicine, South China University of Technology, Guangzhou 510006, China; 2School of Biomedical Sciences and Engineering, Guangzhou International Campus, South China University of Technology, Guangzhou 510180, China; 3National Engineering Research Center for Tissue Restoration and Reconstruction, South China University of Technology, Guangzhou 510180, China; 4Guangdong Provincial Key Laboratory of Biomedical Engineering, South China University of Technology, Guangzhou 510180, China; 5Key Laboratory of Biomedical Materials and Engineering of the Ministry of Education, South China University of Technology, Guangzhou 510180, China

**Keywords:** human embryonic stem cells-derived hepatocytes, hepatic polarized differentiation, hepatocyte polarity, apical membrane

## Abstract

Hepatocytes exhibit a multi-polarized state under the in vivo physiological environment, however, human embryonic stem cell-derived hepatocytes (hEHs) rarely exhibit polarity features in a two-dimensional (2D) condition. Thus, we hypothesized whether the polarized differentiation might enhance the maturity and liver function of hEHs. In this study, we obtained the polarized hEHs (phEHs) by using 2D differentiation in conjunct with employing transwell-based polarized culture. Our results showed that phEHs directionally secreted albumin, urea and bile acids, and afterward, the apical membrane and blood–bile barrier (BBIB) were identified to form in phEHs. Moreover, phEHs exhibited a higher maturity and capacitity of cellular secretory and drug metabolism than those of non-phEHs. Through transcriptome analysis, it was found that the polarized differentiation induced obvious changes in gene expression profiles of cellular adhesion and membrane transport in hEHs. Our further investigation revealed that the activation of Hippo and AMPK signaling pathways made contributions to the regulation of function and cellular polarity in phEHs, further verifying that the liver function of hEHs was closely related with their polarization state. These results not only demonstrated that the polarized differentiation enhanced the maturity and liver function of hEHs, but also identified the molecular targets that regulated the polarization state of hEHs.

## 1. Introduction

For end-stage liver diseases, the only curative treatment is orthotopic liver transplantation (OLT). However, the problems including the insufficient source of the donor livers, appearance of contraindications (such as cardiac or respiratory failure) and adverse effects of posttransplant immunosuppression limit the application of OLT and its therapeutic effect [1]. Nowadays, the hepatocyte transplantation and extracorporeal bioartificial liver devices based on the conception of cell therapy became the effective alternatives to OLT [1,2]. During the practice of liver cell therapies, researchers were eager to obtain fully functional human hepatocytes to achieve the desired therapeutic effect. In addition, the studies associated with human liver physiology, pathology and toxicology also need functional human hepatocytes to establish in vitro models [3,4]. It is no doubt that primary human hepatocytes (PHHs) remain the gold standard for these research fields mentioned above. In recent years, obvious progress has been achieved to maintain long-term functional PHHs [5,6], but the poor availability and stable sources of hPHs still hinder their applications in liver clinical and basic studies [1]. 

Human embryonic stem cells (hESCs) and human-induced pluripotent stem cells (hiPSCs), have wide applications as a robust cell source in regenerative medicine due to their properties of self-renewal and multi-lineage differentiation potentials [7]. Previous studies demonstrated that hiPSCs and hESCs could be differentiated into functional hepatocytes in vitro through mimicking the principal stages of liver development and organogenesis [1,8]. To overcome the PHH shortage, these stem cell-derived hepatocytes have been attempted to use in the studies of liver disease modeling [4], toxicological tests [9], drug screening [10] and bioartificial livers [11]. However, results of those studies exhibited that the maturity of stem cells-derived hepatocytes were only comparable to human fetal/neonatal hepatocytes, and their hepatic function still had a distinct gap with the fully mature adult hepatocytes [12]. Thus, finding an effective way to improve the maturity and liver function of stem cell-derived hepatocytes has attracted extensive attentions in this field.

As specialized epithelial cells in the liver, hepatocytes exhibit a multi-polarized state and possess several apical (canalicular) and basolateral (sinusoidal) domains of the plasma membrane under the in vivo physiological environment [13]. The establishment of hepatocyte polarization phenotype occurs throughout the in vivo liver development, and the loss of hepatic polarity would lead to the occurrence of several liver disorders [14]. Moreover, Zeigerer et al. illustrated that functional properties of PHHs in vitro were correlated with cell polarity maintenance [15]. Unfortunately, stem cell-derived hepatocytes are conventionally differentiated under two-dimensional (2D) culture conditions, and these cells rarely exhibit polarity features. Due to this reason, Palakkan and colleagues managed to polarize hESC-derived hepatocytes by overlaying collagen to form a 2D-matrix sandwich [16]. However, the presence of the collagen overlayer might reduce the differentiation efficiency by hindering growth factors diffusion, and it also made the hepatic cargos excreted into the closed bile canaliculi inaccessible. To overcome the disadvantages of collagen overlayer usage, a recent study described a novel transwell filter-based differentiation protocol that generated columnar polarized stem cell-derived hepatocytes, and their polarized stem cell-derived hepatocytes could recapitulate essential steps of the hepatitis E virus’s natural infections cycle [17]. Although these studies established polarized stem cell-derived hepatocytes, whether polarization differentiation is an effective way to promote the maturity and function of stem cell-derived hepatocytes remained to be further elucidated. 

Our group has already established a stable and efficient differentiation program of functional hEHs with 2D culture [18], yet, the yields of the polarized cells was still at a low level. In this present study, we hypothesized that switching the conventional 2D culture to a transwell-based polarized differentiation culture might further enhance the maturity and hepatic function of hEHs. To prove this hypothesis, we first characterized alterations of the polarity, blood–bile barrier function, maturity and hepatic function in the hEHs. In addition, to reveal the mechanism associated with these changes, followed by the analysis of gene profiles, the phosphorylation level of key proteins during the activation of Hippo and AMPK signaling pathways were also determined in this study. 

## 2. Materials and Methods

### 2.1. Establishment of the Polarized hEHs

The hESC H9 cell line was obtained from the WiCell Research Institute (Madison, WI, USA) under a Materials Transfer Agreement (No. 19-W0512) and inoculated as clonal form on CF-1 mouse embryonic fibroblasts and cultured as per provider instructions. After it was cultured for approximately a week’s time, the hESC clones, which the confluence reached 80%, were used for hepatic differentiation. Definitive endodermal (DE) cells were differentiated as previously described [18]. Briefly, hESC clones were cultured with RPMI 1640 medium supplemented with 100 ng/mL Activin A and 25 ng/mL Wnt3a. Next day, the culture medium was removed and fresh RPMI 1640 medium containing 100 ng/mL Activin A, 0.5 mM sodium butyrate and 1 × B27 was replenished until day 7, then the DE cells were obtained and collected for further differentiation.

To induce polarized hepatic differentiation [19], DE cells were digested by TrypLE Express and re-seeded in homemade Matrigel-coated Transwell with a density of 2000 cells/mm^2^. Then, the cells were cultured in hepatocyte differentiation medium (HDM) consisting of IMDM medium supplemented with 20% FBS, 0.3 mM 1-thioglycerin, 0.5% DMSO, 100 nM dexamethasone, 0.126 U/mL insulin and HDM growth factors cocktail (20 ng/mL FGF-4, 20 ng/mL HGF, 10 ng/mL BMP2 and 10 ng/mL BMP4). In the maturation stage, the cells were further matured in the presence of Hepatocyte basal medium (Lonza, omitting the EGF) supplemented with 0.5% DMSO, 100 nM dexamethasone and HCM growth factors cocktail (20 ng/mL FGF-4, 20 ng/mL HGF and 50 ng/mL Oncostatin M). In order to promote the polarization level of hEHs, from day 10 after the hepatocyte differentiation under HMD condition, HDM medium with fetal bovine serum (FBS) reduced to 10% in the absence of HDM cytokine cocktail was added into the upper chamber while complete HDM with 20% FBS plus HDM growth factors cocktail was added into the lower chamber to induce the polarized differentiation for another 4 days. During the hepatocyte maturation under HCM condition, the HCM basal medium without supplemental kit was added into the upper chamber of transwells while the complete HCM medium (basal medium plus supplemental kit, and EGF omitted) supplemented with HCM growth factors cocktail was added into the lower chamber of transwells to maintain the polarized differentiation. After 5–6 days of HCM stage, the polarized hEHs were ready for downstream analysis and experimental procedures. In order to inhibit the AMPK pathway, the polarized hEHs were treated with 3 μM Compound C (Med Chem Express) for 4 days during the maturation stage with HCM medium culture. Cells were cultured at 37 °C and media were replaced daily. All growth factors were purchased from Peprotech.

### 2.2. Isolation and Culture of Primary Human Hepatocytes 

Human liver tissues were obtained from the excised normal tissues adjacent to the liver tumors provided by the volunteers at Guangzhou First People’s Hospital. The Research Ethical Committee of Guangzhou First People’s Hospital approved this study (Ethical approval No.: K-2019-167). PHHs were isolated from these tissues by two-step collagenase perfusion [20]. Briefly, the tissue was first perfused with pre-warmed Perfusion Medium for 30 min, and then performed with pre-warmed Digestion Medium (including Type I collagenase and Ca^2+^) for 10 min. The digestion was stopped by adding cold DMEM/F12 (Gibco) plus 0.1 % BSA (BioFroxx) to 30 mL in 50 mL centrifuge tube. After being filtered by a 70 μm filter, the PHHs were separated and purified by Percoll gradient centrifugation as described before [21]. The PHHs were plated on a Collagen type I-coated (Corning) culture dish at 4 × 10^5^ cells per cm^2^ and cultured in hepatocyte basal medium (Lonza), and the media were replaced daily. Three days after culture, the PHHs were used for downstream analysis. 

### 2.3. Flow Cytometry Analysis

The cells were digested into single cells using TrypLE Express. Non-specific binding sites on the cells were blocked using mouse serum. After fixing and permeabilizing, the cells were then fluorescently labelled by incubation with PE anti-human SOX17 antibody (BD Biosciences, San Jose, CA, USA) and APC/Cy7 anti-human CXCR4 antibody (BioLegend, San Diego, CA, USA). Fluorescence-positive cells were then detected using a BD FACS Celesta flow cytometer (BD Biosciences). Antibodies used were listed in Appendix A.

### 2.4. Assays for Albumin Secretion, Urea Nitrogen and Total Bile Acids 

The concentrations of albumin (ALB), urea nitrogen and the total bile acids in culture medium were measured by Human Albumin Enzyme Linked Immunosorbent (ELISA) Quantitation Kit (BETHYL), Urea Nitrogen Assay Kit (Sangon Biotech, Shanghai, China) and Total Bile Acid Kits (Nanjing Jiancheng Bioengineering Institute, Nanjing, China) according the manufacturer’s instructions. 

### 2.5. Determination of Cytochrome p450 Activity (CYP3A4) 

For the determination of CYP3A4 enzyme activity, hEHs were induced with 25 μM rifampicin (Sigma-Aldrich, R3501,Saint Louis, USA) for 48 h. CYP3A4 enzyme activities were measured 3 h after the incubation with subtype-specific substrate using a commercial P450-Glo^TM^ Assay Kit (Promega Corporation, Madison, USA). Luciferase activities were then detected by a multi-detection microplate reader, and the results were normalized with cell number. 

### 2.6. Assay for Paracellular Permeability 

With the final concentration of 1 mg/mL, 4 kDa fluorescein isothiocyanate–dextran (Sigma-Aldrich) was added to the upper or lower chamber of Transwell and incubated overnight, respectively. Then, the absorbance of supernatants was detected at 485 nm and 544 nm wavelength using a multi-detection microplate reader, respectively. The paracellular permeability was assayed by calculating the ratio of absorbance between the upper and lower chambers.

### 2.7. Quantitative Real-Time Polymerase Chain Reaction

Total RNAs were extracted from the different cell samples using RNAiso Plus (TAKARA). Following quantification in a Nano Drop micro spectrophotometer (Thermo Fisher Scientific, Waltham, MA, USA ), 1 μg of RNA was converted to cDNA using the PrimeScript™ RT Master Mix (TAKARA). The gene expression was detected using PowerUp^TM^ SYBR^TM^ qPCR Green Master Mix (Thermo Fisher Scientific) with specific primers in a Quant Studio^TM^ 1 Real-Time PCR System (Thermo Fisher Scientific). Relative expression of target genes was represented by normalization to GADPH and calibration to the non-polarized hEHs by using the 2^−ΔΔCt^ method. Primers used were listed in Appendix A.

### 2.8. Western Blot Analysis 

The cells were scraped and sonicated in RIPA lysis buffer containing PMSF and phosphatase inhibitors. The protein concentrations of different sample were measured using the BCA protein quantification kit (Beyotime, Shanghai, China). Boiled protein samples were separated by SDS-PAGE and then electro-transferred onto poly (vinylidene difluoride) PVDF membranes (Merck Millipore, Darmstadt, Germany). After being blocked in 1 × Tris-buffered saline-Tween (TBST) buffer with 2.5% bovine serum albumin (BSA) (15 mM Tris-HCl, 150 mM NaCl, 0.1% Tween 20, and 2.5% BSA, pH 7.4) for at least 60 min at room temperature, PDVF membranes were incubated with primary antibodies. After washing with 1 × TBST, PDVF membranes were incubated with HRP-labeled Goat Anti-Rabbit IgG (H + L) (Beyotime) for 60 min at room temperature. Finally, target bands were visualized using Amersham Imager 600 (GE Healthcare). Intensities of band were measured using the Image J program. Antibodies used were listed in Appendix A.

### 2.9. Immunofluorescence Staining 

The non-polarized and polarized hEHs were fixed using 4% paraformaldehyde fix solution. After permeabilizing by 0.5% Triton and blocking by goat serum, the cells were incubated with primary antibody at 4 °C overnight. Then, the cells were co-incubated with anti-rabbit IgG Fab2 Alexa Fluor (R) 488 (Cell Signaling Technology, Boston, MA, USA) and Phalloidin-iFluor 594 (Abcam, Cambridge, UK). After washing, the cells were incubated with DAPI staining solution (Beyotime). After being thoroughly washed, the samples were removed from the hanging insert and placed on microscope slides. The fluorescence signal was imaged on the single photon confocal microscopy (Ti-E A1, Nikon, Tokyo Met. Japan). Antibodies used were listed in Appendix A.

### 2.10. RNA Sequencing

Total RNAs of the non-polarized and polarized hEHs were extracted using RNAiso Plus. The concentration of RNA was accurately measured by Qubit fluorometer, and the integrity of the RNA was accurately detected using Agilent 2100 BioAnalyzer. The RNA sequencing libraries were constructed using NEBNe Ultra^TM^ RNA Library Prep Kit for Illumina (San Diego, CA, USA). The concentration of the libraries was measured and diluted into 1 ng/μL. The insert size of the libraries was then measured using an Agilent 2100 BioAnalyzer. Then, the effective molar concentration of the libraries was absolutely quantified by qRT-PCR to ensure that it was greater than 2 nM. After that, different libraries were sequenced on Illumina HiSeq X-Tensequencer with 150 bp paired-end sequencing reaction. Sequencing was performed by Novogene (Beijing, China). Raw reads were firstly obtained by sequencing, and clean reads were obtained after filtration. The clean reads were mapped to the human reference genome for comparison using HISAT2, and the readings of each gene in each sample were counted with FeatureCounts. Then, FPKM (Fregments Per Kilobase per Million) was calculated according to the length of the gene to estimate the gene expression level of each sample. DESeq2 R software was used to analyse the differentially expressed genes (DEGs) between the experimental group and control group, and the Benjamini or Hochberg method was used to adjust the *p* values obtained to control the false discovery rate. Genes with *p* value ≤ 0.05 and log2 Fold Change ≥ 1 was identified as DEGs. Heatmap generation was performed with the R package. TBtools software and KOBAS database were used to test the statistical enrichment of DEGs in KEGG pathways and Gene Ontology. Additionally, Gene set enrichment analysis (GSEA) was conducted by MSigDB database and GSEA software.

### 2.11. Statistics

Graphs and statistical analyses were performed using GraphPad Prism 8.0 (GraphPad Software, San Diego, CA, USA). Data was expressed as the mean ± standard deviation (n = 3). Statistical significance was determined by an unpaired two-tailed Student’s *t* Tests. Differences were considered statistically significant where a *p* value < 0.05.

## 3. Results

### 3.1. Generation of the Polarized hEHs

Before the DE cells were seeded in transwell chambers, we evaluated the differentiation efficiency using flow cytometry. As shown in Figure 1B, flow cytometry analysis revealed that the positive populations for DE cell markers C-X-C Motif Chemokine Receptor 4 (CXCR4) and sex-determining region Y-box 17 (SOX17) were over 90%. Moreover, sequential morphological alterations were observed both in the non-polarized and polarized hEHs during the hepatic differentiation (Figure 1C–E). As shown in Figure 1E and Figure 2A, the polarized differentiation procedures generated hEHs (the polarized hEHs) with a higher ratio of binucleate than those of 2D cultured hEHs (the non-polarized hEHs). Both the polarized and non-polarized hEHs exhibited time-dependent expression of hepatic markers (*ALB*, *G6P, α1-AT*, *CK18* and *CK19*), which are expressed in fetal and adult hepatocytes (Figure 1F and Appendix A). In addition, compared to DE cells, the expression of *SOX17* gene was sharply decreased, while the expression of hepatic markers (*ALB*, *G6P*, *α1-AT* and *CK19*) was obviously increased in both the non-polarized and polarized hEHs during the hepatic differentiation (Figure 1F and Appendix A). These results indicated that the newly developed polarized differentiation conditions did not affect the differentiation efficiency of hEHs.

### 3.2. The Polarized hEHs Directionally Secreted the Plasma Proteins, Urea and Bile Acids

As shown in Figure 2A, the polarized differentiation condition generated more binucleate hEHs which are the hallmark of mature hepatocytes. As an in vivo secretion factory, hepatocytes directionally secrete cargos, in which plasma proteins are mainly secreted to the sinusoid, while bile acids are mainly secreted to bile canaliculus [22]. Synthesized only by hepatocytes, ALB is a key functioning protein secreted into blood circulation. We thus measured the dynamic levels of secreted ALB in the mediums of upper and lower chambers during the period of the polarized differentiation. The hEHs began to secrete the majority of the total ALB to the lower chambers after the induction of the polarized differentiation, and the total ALB secretion reached a peak at day 27 (Figure 2B). In line with the direction of ALB secretion, over 70 % of the total urea (the ammonia metabolite of hepatocytes) were also secreted into the lower chambers (Figure 2C,D). On the contrary, in the opposite direction of ALB secretion, approximately 75% of the total bile acids were secreted into the upper chambers of the transwells (Figure 2C,E). These results demonstrated that the polarized hEHs could directionally secrete plasma proteins and bile acids. In addition, in order to verify this directional secretion, we assessed the paracellular permeability of the polarized hEHs in the transwells using the 4 kDa FITC-labeled dextran. As shown in Figure 2F, whatever FITC-labeled dextran was placed in the upper chambers or the lower chambers, FITC-fluorescence signals were not detected in the opposite chambers of transwells in the presence of the polarized hEHs, whereas either the upper or the lower chambers had equal fluorescence signals in the absence of the polarized hEHs, indicating such directional secretion was not caused by the paracellular permeability. As a result, this phenomenon demonstrated that the directional secretion of hepatic cargos was due to the active transport of the polarized hEHs rather than their paracellular diffusion.

### 3.3. Identification of Apical Membrane Proteins and BBIB in the Polarized hEHs

It is well documented that the directional secretion of hepatocytes relies on the appearance of cellular apical and basal membranes [22]. The bile salt export pump (BSEP) is a member of the ABC transporters, which is located on the hepatic apical membrane, and it is responsible for the excretion of intracellular bile acids into the bile canaliculus [17]. Thus, we analyzed the differences in expression and distribution of BSEP between the non-polarized and polarized hEHs. The protein level of BSEP was analyzed by Western blot, and we found a significant up-regulation of BSEP protein expression in the polarized hEHs when compared to the non-polarized hEHs (Figure 3A). Meanwhile, the multi-drug resistance protein 1 (MDR1), another marker of the hepatic apical membrane [22], was also significantly increased in the polarized hEHs when compared to those of the non-polarized hEHs (Figure 3B; *p* < 0.05). In addition, we also quantified the organic aniontransporting polypeptide 1 (OATP1), a hepatic basal marker in this study. Our result showed that the gene expression of OATP1 was significantly decreased in the polarized hEHs than that of the non-polarized hEHs. To determine the pattern of cellular polarity, we performed co-immunostaining of hEHs with F-actin and BSEP (relatively concentrated on the canalicular membrane of hepatocytes in the human liver tissue) and analyzed their z-stack confocal images. The results of the confocal images showed that F-actin and BSEP were mainly co-localized on the upper (apical) membrane in the polarized hEHs, while most of BSEP were localized with the nucleus and F-actin were distributed in the lateral membrane of the cell in the non-polarized hEHs (Figure 3C). These observations provided strong evidence that the polarized hEHs formed a typical apical membrane domain and changed the gene expression levels of basal membrane proteins more than the non-polarized hEHs did.

A previous study pointed out that hepatic tight junctions (TJs) not only established the BBIB that segregates bile from the blood circulation, but also reinforced cell polarity by maintaining the segregation of hepatic apical and basal membranes [23]. To confirm whether the polarized hEHs could formed BBIB, we determined the distribution and expression of TJ proteins (ZO-1 and E-cadherin) between the non-polarized and polarized hEHs. As observed by confocal analysis of ZO-1 immunostaining, we found a strong fluorescent intensity increase of ZO-1 in the polarized hEHs when compared with the non-polarized hEHs (Figure 3D). Meanwhile, our observations also showed that the distribution of ZO-1 was mainly localized in the polarized hEHs boundaries, and tightly arranged with sharp, continuous smooth edges, and without abnormalities. In contrast, only a few fractions of ZO-1 fluorescence signals were detected at the cellular edges and rarely formed a network in the non-polarized hEHs (Figure 3D). In addition, Zhang et al. proved that E-cadherin played a key role in triggering the emergent organization of hepatic polarized phenotype [24]; we thus found the protein expression levels of E-cadherin were significantly increased in the polarized hEHs than those of the non-polarized hEHs (Figure 3A; *p* < 0.05). Overall, these data strengthened the fact that the polarized hEHs maintained the polarity statue by BBIB formation.

### 3.4. The Polarized Differentiation Improved the Maturation and Liver Function of hEHs

The above results illustrated that the polarized hEHs could be obtained through polarized differentiation, yet whether this differentiation culture influenced the maturation and liver function of hEHs was still not clear. Hence, we first compared the differences in hepatic maturity of the non-polarized hEHs, polarized hEHs, PHH and DE cells. After hepatic differentiation, the polarized hEHs had a higher gene expression level of mature hepatic markers (*ALB*, *G-6-P* and *CPS1*) and a lower gene expression of the immature hepatocyte markers (*AFP* and *CK19*) than those of the non-polarized hEHs (Figure 4A–E). In addition, the polarized hEHs also showed a higher level of ALB secretion compared with the non-polarized hEHs (Figure 4F). The secretion of functioning proteins is one of the major properties of mature hepatocytes, thus, these results indicated that the polarization differentiation promoted the hepatic maturity of hEHs. In line with the increase of gene expression of *CPS1* (the rate-limiting enzyme of urea synthesis), the urea secretion level was significantly increased in the polarized hEHs (Figure 4C,G; *p* < 0.05), indicating that the polarized differentiation enhanced both the urea synthesis and secretion capacity of hEHs. Notably, we also found that the polarized hEHs had a higher level of bile acid secretion than that of the non-polarized hEHs (Figure 4H). In an attempt to further validate the effect of the polarized differentiation culture on the capacity of drug metabolism, we treated the polarized hEHs from their basolateral side (lower chamber) with rifampicin (Rif), one of the CYP3A4 inducers. Forty-eight hours after exposure to Rif, the enzymatic activities of CYP3A4 in the polarized hEHs was significantly higher than that of the non-polarized hEHs (Figure 4I); an important property of mature hepatocytes is to respond to inducers. Thus, these findings demonstrated that the polarized differentiation profoundly increased the capacity of drug metabolism in hEHs.

Moreover, compared with the non-polarized hEHs, the gene expressions of *FXR* and *SHP* were significantly up-regulated, while the gene expression of CYP7A1, bile acids’ rate-limiting enzyme which is regulated by FXR, was down-regulated in the polarized hEHs (Figure 4K–M). These results suggested that the polarized differentiation culture not only promoted the bile acid secretion, but also made hEHs maintain a typical bile acid feedback regulation through the FXR-SHP-CYP7A1 signaling axis (Appendix A). Rif as the most commonly used first-line drugs in antituberculosis therapy, has been well known to be hepatotoxic [25]. Previous studies confirmed that Rif-induced cholestatic liver injury was characterized by impaired hepatic apical transports such as BSEP and MRP2 [26,27]. To test if the differentiated cells could mimic phenotypes of Rif-induced liver injury in vitro, we compared the polarized hEHs with the non-polarized hEHs for their ability to express bile acid transporters that respond the Rif exposure. As expected, the gene expression of *BSEP* and *MRP2* were down-regulated in the polarized hEHs treated with Rif (Figure 4N,O), thus, consistent with these results, the total bile acid secretion was also decreased in the polarized hEHs after the treatment with Rif (Figure 4P). Meanwhile, with the down-regulation of the apical bile acid transporters, the polarized hEHs lost the capacity of directional secretion of bile acids (Figure 4J). In addition, a recent in vivo study revealed that Rif reduced hepatic bile acid levels by suppressing the *FXR* gene expression; in agreement with their results, we thus found the treatment with Rif significantly decreased the *FXR* gene expression in the polarized hEHs (Figure 4K). Furthermore, although *FXR* gene expression was decreased, the treatment with Rif increased the *SHP* gene expression and did not affect the *CYP7A1* gene expression (Figure 4L,M), indicating that the Rif might influence the bile acid metabolism through disturbing the FXR-SHP signaling axis. Notably, these changes were not observed in the non-polarized hEHs treated with Rif (Figure 4K–P). Altogether, these findings highlighted that the polarized hEHs could mimic key phenotype of drug-induced liver injury similar to in vivo pathological progress (Appendix A). 

### 3.5. The Polarized Differentiation Altered the Gene Expression Profiles of hEHs

Although the alterations of several key gene expressions confirmed that the hepatic maturity and liver function of the polarized hEHs were enhanced, we were still encouraged to fully reveal the effect of the polarization differentiation culture on the gene expression profiles of hEHs. Thus, RNA-seq analysis was performed to assess the differences between the non-polarized and polarized hEHs. Heatmaps cluster analysis showed that a total of 1025 differentially expressed genes (DEGs) were enriched between the non-polarized and polarized hEHs (Figure 5A). When compared with the non-polarized hEHs, the volcano maps exhibited that 653 differential genes were significantly up-regulated, and the other 372 genes were significantly down-regulated in the polarized hEHs (Figure 5B). These results indicated that the polarization differentiation culture could distinctly alter the gene expression profiles of hEHs. Based on the enrichment of DEGs, we found that the genes associated with the cell adhesion (*CEACAM1* and *EPCAM*), the formation of cellular apical membrane microvilli (*CDHR2*), the desmosome assembly (*DSC2*, *DSG2* and *PERP*), the apical membrane of ABC transporters (*ABCB11*, *ABCC2* and *ABCG2*) and the membrane solute carriers (*SLCs*) were up-regulated and enriched in the polarized hEHs (Figure 5C,D). Notably, consistent with the increase of CYP3A4 enzyme activity mentioned above, we selected the genes of phase I drug-metabolizing enzymes (CYP450) and phase II drug-metabolizing enzymes (UGTs) for heatmap clustering analysis. As shown in Figure 5E,F, most of the CYP450 and UGT expressions were up-regulated in the polarized hEHs than those of the non-polarized hEHs. In addition to the heat map clustering analysis, the gene ontology (GO) analysis was also performed using the DEGs. As shown in Figure 5G, the microvillus membrane, basal part of cell, apical part of cell, golgi lumen and endoplasmic reticulum lumen were mainly enriched in the class “cellular component”, while the cell–cell adhesion, organic acid transport, lipid transport and fatty acid metabolic process were enriched in the class “biological process”. These results indicated that the polarization differentiation culture obviously induced the changes of cellular adhesion and membrane transport.

### 3.6. Hippo Signaling Pathway Was Activated in the Polarized hEHs

Based on the above findings, we revealed that the cellular maturity and liver function of hEHs were dramatically improved under the polarized differentiation, which prompted us to further explore the underlying molecular mechanism. After performing kyoto encyclopedia of genes and genomes (KEGG) pathway analysis using the identified DEGs, we found that the MAPK signaling pathway, the PPAR signaling pathway, the Wnt signaling pathway and the Hippo signaling pathway were significantly enriched (Figure 6A). Notably, a previous study has well documented that the activation of Hippo signaling is essential for promoting the maturity and functional differentiation of induced hepatocyte-like cells [28]. Hence, we examined the phosphorylation and cellular localization of Yap, which is an important downstream effector molecule of the Hippo signaling pathway to determine whether the Hippo signaling pathway was activated in the polarized hEHs (Figure 6B). As expected, the phosphorylation level of YAP protein was significantly increased in the polarized hEHs (Figure 6C). Meanwhile, immunostaining results further confirmed that most of Yap was located in the cytoplasm of the polarized hEHs, while a high percentage of Yap was translocated into the nucleus of the non-polarized hEHs, indicating the activation of the Hippo signaling in the polarized hEHs (Figure 6E). In addition, in line with the inhibition of translocation of Yap into the nucleus, we also found that the gene expression level of Yap downstream genes (*Ctgf* and *Cyr61*) was sharply decreased in the polarized hEHs than those of the non-polarized hEHs (Figure 6D). These results indicated that the polarization differentiation enhanced the maturity and liver function of hEHs via activating Hippo signaling pathway.

### 3.7. Activation of AMPK Signaling Pathway Was Involved in the Maintenance of the Polarity in the Polarized hEHs

Although the polarized hEHs were characterized, which signaling mechanisms were involved in the maintenance of the polarity in the polarized hEHs needed further elucidation. Previous studies confirmed that the AMP-dependent kinase (AMPK) played an important role in bioenergetics for hepatocyte polarization [29]. Accordingly, we speculated that the AMPK signaling pathway mediated the maintenance of the polarity in our polarized hEHs. To verify this possibility, we collected the polarized hEHs and treated them with Compound C (CC), a specific inhibitor of AMPK for 4 days during the maturation stage with HCM medium culture, and Western blotting was performed to measure the phosphorylation status of AMPK in the polarized hEHs (Appendix A). As expected, corresponding to the up-regulation of BSEP and NTCP protein expressions, the polarized differentiation induced an increased level of AMPK phosphorylation, and these effects were obviously inhibited by CC (Figure 7A). We also examined the phosphorylation status of YAP in the polarized hEHs in the presence of CC, and it was found that the incubation of CC for four days led to a significant decrease of YAP phosphorylation in the polarized hEHs (Figure 7A). In addition, the capacity of the directional secretion of bile acids in the polarized hEHs was abolished by the presence of CC (Figure 7B). Moreover, we also observed that the CC treatment decreased the total secretion of bile acids in the polarized hEHs (Figure 7C). In order to verify whether these changes were caused by the effect of the treatment with CC on the polarized differentiation, we found that the capacity of the directional ALB secretion and the total ALB secretion were not changed in the polarized hEHs during the exposure to CC, which indicated the treatment with CC did not alter hepatic differentiation of hEHs except the cellular polarity (Figure 7D–F). All these results demonstrated that the activation of the AMPK signaling pathway mediated the maintenance of the polarity in the polarized hEHs (Figure 7G).

## 4. Discussion

As the parenchymal cells in the liver, hepatocytes produce 90% of the total hepatic proteins and are mainly responsible for bile production, drug metabolism and detoxification [30]. Similar to other epithelial cells in vivo, hepatocytes must be polarized to be functional, which implies the formation of apical and basolateral membranes [14]. Although most epithelial cells typically exhibit columnar polarity, hepatocytes have a unique polarization arrangement which is organized in the plates in the 3D environment of the liver and have several apical and basolateral membranes [14]. In this present study, we explored the effect of transwell-based polarized differentiation culture on the polarity, maturity and liver function in hEHs using our previous differentiation program. As a result, we successfully obtained optimal polarized hEHs (columnar polarization) through using a polarized differentiation protocol. Consistent with reported findings, our polarized hEHs mainly formed columnar polarization on the transwells [17]. Compared to our conventional 2D protocol, the polarization differentiation enhanced the liver function of hEHs in the following aspects, including forming an apical membrane domain that could be characterized by bile canalicular membrane proteins and the directional secretions of ALB, urea and bile acids, establishing a polarized network of cells like the physiological state in vivo, and promoting the capacities of cellular secretory (such as ALB and bile acids) and drug metabolism. According to these changes of cellular function, the polarized hEHs showed higher maturity than the non-polarized hEHs, which indicated the polarization differentiation protocol successfully matched our expectation. In agreement with our findings, the reasons why the collagen sandwich culture system and 3D organoid technology can improve the maturity and liver function of stem cell-derived hepatocytes are also closely related to the shaping of cellular polarization [10,16]. Although no studies have revealed why hEHs can obtain polarized states on transwells, we speculate that two factors are closely involved in this process, one factor is that Matrigel-coated Transwell provides a moderate stiffness for cell adhesion, and defines the basement membrane of cells, and the other is that the growth factor concentrations of the canalicular side is lower than those of the sinusoidal side, thus the generation of growth factor gradients in the side of the apical and basel membranes is also contributed to the establishment and maintenance of cell polarization. All these findings confirmed that the maturity and liver function of hEHs is closely related with their polarization level. 

The BBlB promotes many essential liver functions including generation and maintenance of hepatocyte polarity and regulation of bile acid secretion [23]. TJs are primarily responsible for the barrier properties of the BBlB, and they function to seal the paracellular spaces between hepatocytes [14]. As an integral component of TJs, ZO-1s interact with different TJ proteins and anchor themselves to the actin cytoskeleton and play a key role in assembly and maintenance of TJ integrity [23]. In agreement with these views, we observed a continuous and complete ZO-1 network in the polarized hEHs when compared with the non-polarized hEHs, indicating that the polarized hEHs maintained the polarization state by forming BBIB. In addition, F-actin also made a contribution to BBlB formation by providing anchorage of the TJ proteins to the adaptor molecules, and they were mainly distributed along the plasma membrane of hepatocytes, concentrating at the apical membrane domain [23]. Recently, Zhang et al. demonstrated that the apical lumen development of hepatocytes mainly depended on actin cortex arrangements at the single-cell level [24]. Consistently, our observations also showed that the distribution of F-actin was mainly concentrated at cellular boundaries and colocalized with ZO-1 and BSEP in the polarized hEHs, whereas most fractions of F-actin fluorescence signals were detected in the cytoplasm of the non-polarized hEHs. Notably, in line with the BBIB formation, we also found the focal adhesion, regulation of actin cytoskeleton and cell adhesion molecules signaling pathways were obviously enriched after performing KEGG pathway analysis using the identified DEGs. Taken together, these findings demonstrated that the polarized hEHs were generated and maintained with the cellular polarity by forming BBIB.

Drug-induced liver injury (DILI) is a common adverse drug reaction, and it can lead to liver failure and even death [31]. For the pharmaceutical industry, approximately thirty percent of drugs were withdrawn from the market due to inefficiency of physiologically relevant preclinical models in evaluating DILI in human [10]. Stem cell-derived hepatocytes in vitro models have been shown effective in modeling different types of DILI, including phospholipidosis and steatosis [32,33]. However, these cells express significantly lower levels of CYP enzymes and apical transporters (such as BSEP and MRP2) compared to hPHs [32]. Notably, our present study demonstrated that the polarized differentiation profoundly increased the gene expression of CYP enzymes and apical transporters in hEHs. Interestingly, utilizing our polarized hEHs could more closely simulate in vivo pathological progress of Rif-induced DILI, such as inhibiting the gene expression of *BSEP* and *MRP2* and disturbing the bile acid metabolism. Recently, Shinozawa et al. reported a high-throughput system to measure bile transport activity by stem cell-derived hepatocyte organoids in the presence of testing compounds [10]. Compared with their findings, the ability of cells to excrete drug-derived metabolites from apical membrane made our polarized hEHs have great potentials in applications for studying drug efflux and drug–drug interactions in DILI.

Previous research revealed that the improvement of maturation and function of hEHs was associated with the activation or inactivation of different signaling pathways [34]. Our previous study revealed that activation of Wnt pathway and inhibition of Notch pathway enhanced hepatic differentiation of hHEs [35,36]. In addition, Gao et al. demonstrated that inhibiting the MAPK/ERK signaling pathway, and subsequently attenuating the WNT signaling pathway negatively regulated hepatic differentiation of hESC-derived hepatic progenitors [37]. In agreement with these results, we found that the MAPK and Wnt signaling pathways were remarkably enriched based on the RNA-seq data in this study. These results indicated that hepatic differentiation was a complex process, and the polarization differentiation culture could extensively affect the different signaling pathways during the polarized differentiation. Especially, the Hippo signaling was also enriched in this present study. The Hippo signaling pathway is an evolutionarily conserved signaling module, and it is closely related to liver size control, regeneration, development and tumorigenesis [38]. In particular, the many components of the Hippo pathway are apically localized, which is important for their activity [39]. Borreguero-Muñoz N et al. confirmed that the Hippo pathway had a physiological function as an integrator of epithelial cell polarity, tissue mechanics, and nutritional cues to control cell proliferation and tissue growth in both *Drosophila* and mammals [40]. Thus, we particularly chose the Hippo signaling pathway for verification in this study. Recently, Alder et al. suggested that the Hippo signaling pathway might affect hepatocyte differentiation by influencing HNF4A and FOXA2 interactions with temporal enhancers [41]. Another study from Yamamoto and coworkers showed that cell-aggregate formation could rapidly induce growth arrest and hepatic maturation of stem cell-derived hepatocytes through the activation of Hippo signaling [28]. These studies indicated that the Hippo signaling was involved in regulating cellular maturity and function in hepatic differentiation. Consistent with their findings, we also found that the phosphorylation level of YAP, a key major executive protein of the Hippo signaling, was increased, and most of YAP were located in the cytoplasm of the polarized hEHs, indicating the activation of the Hippo signaling in the polarized hEHs. Interestingly, previous research pointed out that either adherent junctions or basolateral polarity complexes could independently regulate the Hippo pathway in *Drosophila* and mammalian cell models [42]. Based on the conservation of the Hippo pathway in *Drosophila* and vertebrates [38], it appeared that the establishment of cellular polarity was coupled with activating Hippo signaling, which regulated cell maturity and liver function in the polarized hEHs. Thus, according to these findings, we concluded that the polarization differentiation culture could extensively affect the different signaling pathways during the hepatic differentiation. 

Kang et al. demonstrated that energy production was also a key evolutionarily conserved element of hepatocyte polarization formation [29]. AMPK, a serine threonine kinase, controls energy metabolism within cells by sensing the cellular AMP to ATP ratio [14]. It is widely known that the activation of AMPK by phosphorylation of the alpha subunit Thr172 decreases energy consumption and increases energy production. Thus, AMPK has an important role in hepatic metabolism through effects on glucose, lipid and protein homeostasis and mitochondrial biogenesis [14,43]. Notably, one study showed that AMPK regulated canalicular network formation and maintenance in rat primary hepatocytes [44]. Another study reported that the bile acids stimulated hepatocyte polarization through the activation of AMPK signaling [45]. In short, it has been shown that AMPK acts as the master regulator for hepatocyte polarization. In line with these findings, we found the AMPK signaling was activated in the polarized hEHs when compared with the non-polarized hEHs. These results indicated the polarization formation following with the AMPK activation and energy production in the hEHs. Previous studies confirmed that the activation of AMPK facilitated TJ assembly in Caco-2 and MDCK cell lines [46,47]. According to these findings, we logically speculated that the activation of AMPK signaling might maintain the cellular polarity in the polarized hEHs through regulating TJ assembly. Interestingly, our results also verified that both the Hippo pathway and AMP-activated protein kinase (AMPK) were activated during the polarization differentiation of hEHs. Here, these findings demonstrated that the activation of AMPK signaling pathway made a remarkable contribution to generate and maintain the cellular polarity in the polarized hEHs.

## 5. Conclusions

In conclusion, we obtained the polarized hEHs through using the transwell filter-based differentiation protocol in this study, and these polarized hEHs showed a higher maturity and liver function than those derived from the 2D cultured condition. In addition, it was also revealed the activation of Hippo and AMPK signaling pathways were involved in the function improvement and the polarity maintenance of the polarized hEHs, respectively. These results not only demonstrated that the liver function of hEHs was closely related with their polarization level, but also identified the molecular targets that regulate the polarization level of hEHs. 

## Figures and Tables

**Figure 1 cells-11-04117-f001:**
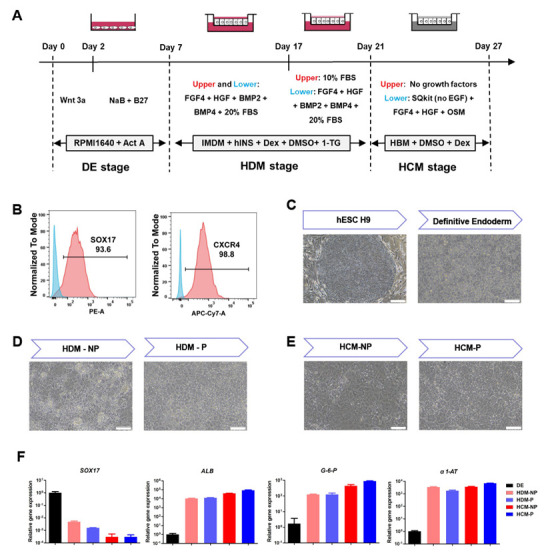
Generation of the polarized hEHs through the transwell−based polarized differentiation (**A**) Schematic representation showing the differentiation protocol to generate the polarized hEHs. (**B**) Flow cytometry determined the proportion of SOX17^+^ and CXCR4^+^ cells in DE cells. (**C**–**E**) Sequential morphological alterations of the nonp−olarized and polarized hEHs during the hepatic differentiations. (**F**) Quantitative PCR analysis of DE cell markers and hepatic−lineage markers in DE cells, the non-polarized hEHs and polarized hEHs cultured with HDM medium, the non−polarized hEHs and polarized hEHs cultured with HCM medium. The aforementioned cells were sampled for quantitative PCR analysis at days 7, 21 and 27 during the hepatic differentiation. Relative gene expression represents data normalized to GADPH and calibrated to DE cells. Data represent the mean ± SD.

**Figure 2 cells-11-04117-f002:**
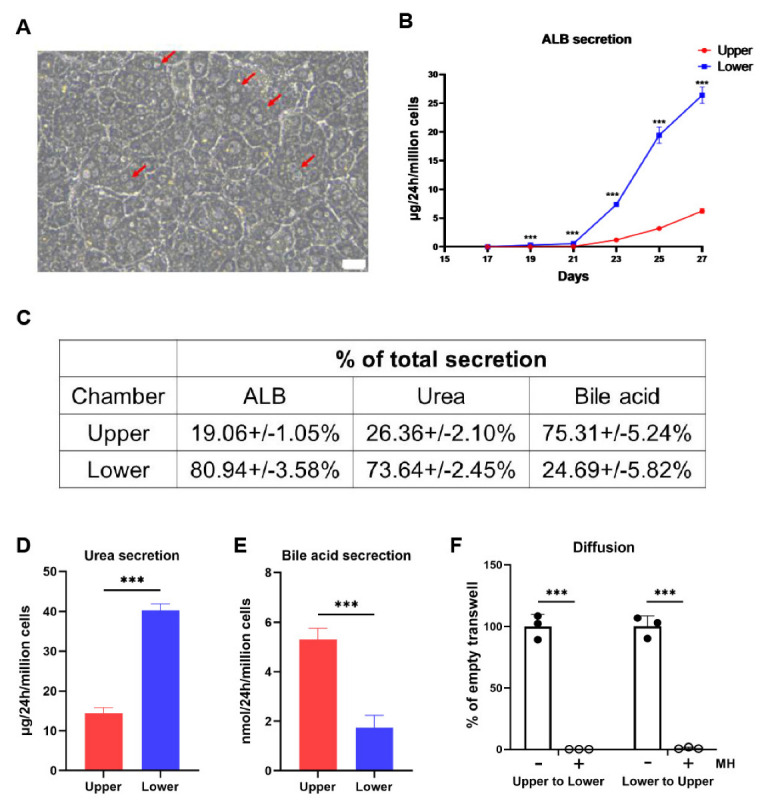
Characterization of the capacity of the directional secretions in the polarized hEHs. (**A**) Morphology of the polarized hEHs at day 27 after the differentiation, red arrows indicated binucleated polarized hEHs, scale bars = 20 μm. (**B**) Time course of the directional ALB secretion in the polarized hEHs through ELISA analysis. (**C**) Calculation of the secretion ratio of hepatic cargos in the medium between the upper and lower chambers. (**D**,**E**) Analysis of the directional secretions of urea and bile acids in the medium of the polarized hEHs between the upper and lower chambers at day 27 after the differentiation. (**F**) The determination of the paracellular permeability of the polarized hEHs incubated with 4kDa FITC−dextran by fluorescence analysis. The symbols “+” and “−” represent the presence or the absence of the polarized hEHs, fluorescence signals of FITC−labeled dextran that were not detected in the opposite chamber of transwells in the presence of the polarized hEHs, whereas either the upper or lower chambers had equal fluorescence signals in the absence of the polarized hEHs, indicating the directional secretions by the polarized hEHs were not caused by the paracellular permeability. Data represent the mean ± SD. *** *p*  <  0.001.

**Figure 3 cells-11-04117-f003:**
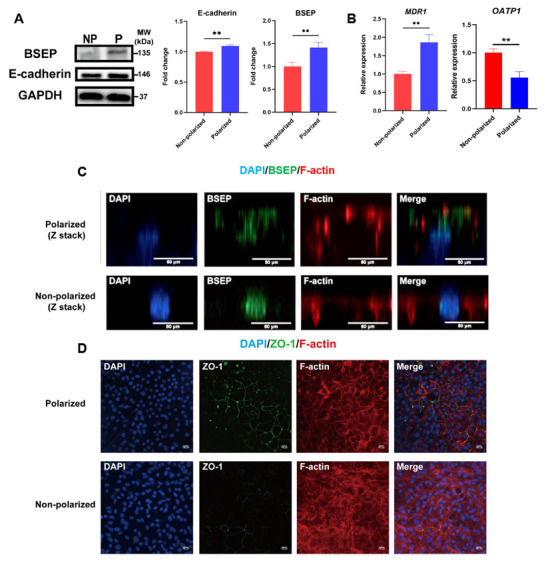
Identification of apical membrane proteins and ZO-1 in the polarized hEHs. (**A**) Western blot analysis of cell lysates from the non-polarized and polarized hEHs for the detection of BSEP and E-cadherin protein expression level (left panel), quantification of these protein levels normalized to GAPDH protein (right panel). Full-length blots were presented in Appendix A. (**B**) Quantitative PCR analysis of apical and basal transporter gene (MDR1 and OATP1) in the non-polarized and polarized hEHs. Relative gene expression represented data normalized to GADPH and calibrated to the non-polarized hEHs. (**C**) Immunofluorescence analysis of BSEP (green) and F-actin (red) on the non-polarized and polarized hEHs; the nuclei were stained with DAPI (blue). Scale bars = 50 μm. (**D**) Immunofluorescence analysis of ZO-1 (green) and F-actin (red) on the non-polarized and polarized hEHs; the nuclei were stained with DAPI (blue). Scale bars = 100 μm. Data represent the mean ± SD. * *p*  <  0.05, ** *p*  <  0.01, and *** *p*  <  0.001.

**Figure 4 cells-11-04117-f004:**
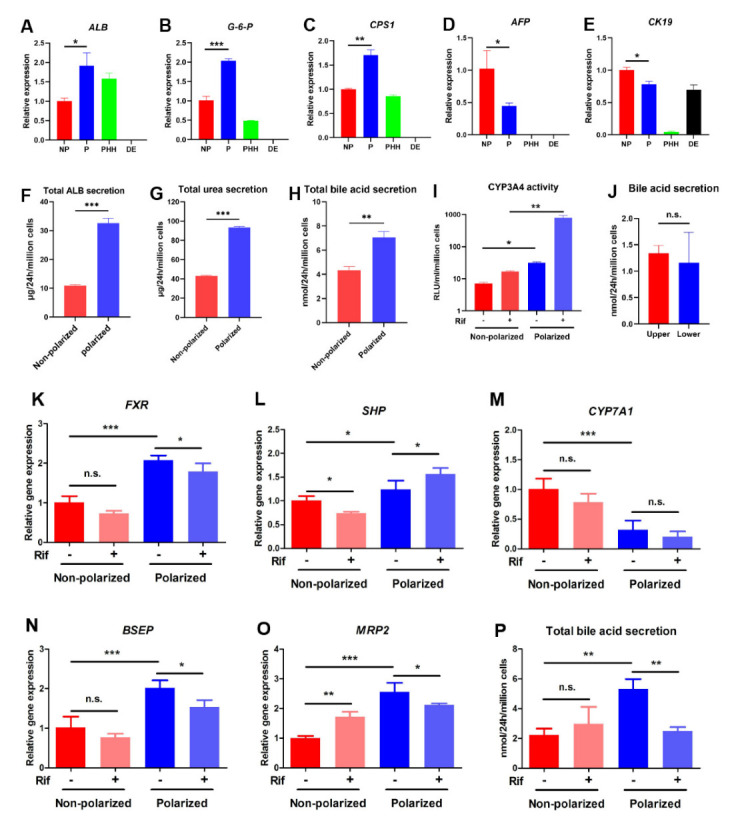
The effects of polarized differentiation on the maturation and function of hEHs. (**A**–**E**) Quantitative PCR analysis of hepatic maturation-related genes (*ALB*, *G−6−P*, *CPS1*, *AFP* and *CK19*) in the non-polarized and the polarized hEHs. Relative gene expressions were normalized to GADPH and calibrated to the non-polarized hEHs. (**F**) ELISA analysis of the total ALB secretion in the non-polarized and polarized hEHs at day 27 after the differentiation. (**G**) Analysis of the total urea secretion by commercial Urea Nitrogen Assay Kit in the non-polarized and polarized hEHs at day 27 after the differentiation. (**H**) Analysis of the total bile acid secretion of the non-polarized and polarized hEHs by using Total Bile Acid Kits. (**I**) Assessment of CYP3A4 activity of the non-polarized and polarized hEHs by commercial P450−GloTM Assay Kit in the presence of Rif for 2 days using fluorescence-based assays. (**J**) Analysis of the directional secretion of the bile acids in the polarized hEHs by using Total Bile Acid Kits treated with Rif for 2 days. (**K**–**O**) Quantitative PCR analysis of hepatic bile acid metabolism related genes and apical transporters of the non-polarized and polarized hEHs in the presence or the absence of Rif for 2 days. Relative gene expression represented data normalized to GADPH and calibrated to the non-polarized hEHs without the induction of Rif. (**P**) Analysis of the total bile acid secretion of the non-polarized and polarized hEHs by using Total Bile Acid Kits in the presence or the absence of Rif for 2 days. Data represent the mean ± SD. * *p* <  0.05, ** *p * <  0.01, *** *p*  <  0.001; n.s. means no significant.

**Figure 5 cells-11-04117-f005:**
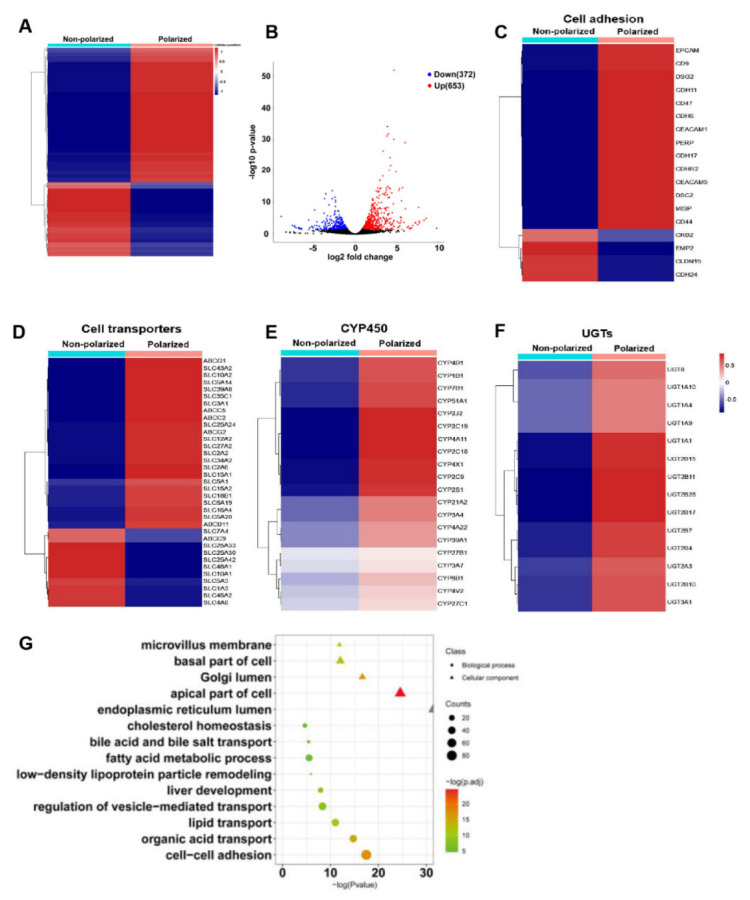
The effect of polarized differentiation on the transcriptome profiles of hEHs (**A**) Heatmap of DEGs of hEHs, the non−polarized vs. the polarized hEHs. (**B**) Volcano map of DEGs in the non−polarized v.s. the polarized hEHs. Red dots represented significantly upregulated DEGs, blue dots represented significantly downregulated DEGs and grey dots represented no significance between them. (**C**) Heatmap of cell adhesion related genes in the non−polarized v.s. the polarized hEHs. (**D**) Heatmap of cell transporter related genes in the non−polarized v.s. the polarized hEHs. (**E**) Heatmap of CYP450 related genes in non−polarized v.s. polarized hEHs. (**F**) Heatmap of UGT related genes in non−polarized vs. polarized hEHs. (**G**) GO terms of DEGs of hEHs, the non−polarized vs. the polarized hEHs.

**Figure 6 cells-11-04117-f006:**
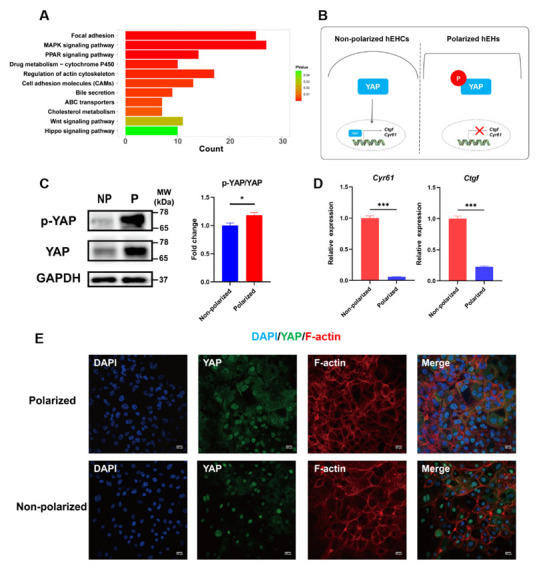
Polarized differentiation activated Hippo signaling pathway in hEHs. (**A**) KEGG pathway enrichment of hEHs between the non−polarized and polarized hEHs. (**B**) Schematic representation depicting the activation of Hippo signaling pathway by the phosphorylation of the YAP in the polarized hEHs. (**C**) Western blot analysis of cell lysates from the non-polarized and polarized hEHs determining YAP phosphorylation level (left panel), quantification of these protein levels normalized to GAPDH protein (right panel). Full−length blots were presented in Appendix A. (**D**) Quantitative PCR analysis of YAP downstream genes in the non-polarized and polarized hEHs. Relative gene expression represented data normalized to GADPH and calibrated to the non−polarized hEHs. (**E**) Immunofluorescence analysis of YAP (green) and F−actin (red) in the non−polarized and polarized hEHs; the nuclei were stained with DAPI (blue). Scale bars = 100 μm. Data represent the mean ± SD. * *p* <  0.05, ** *p*  <  0.01, *** *p*  <  0.001; n.s. means no significant.

**Figure 7 cells-11-04117-f007:**
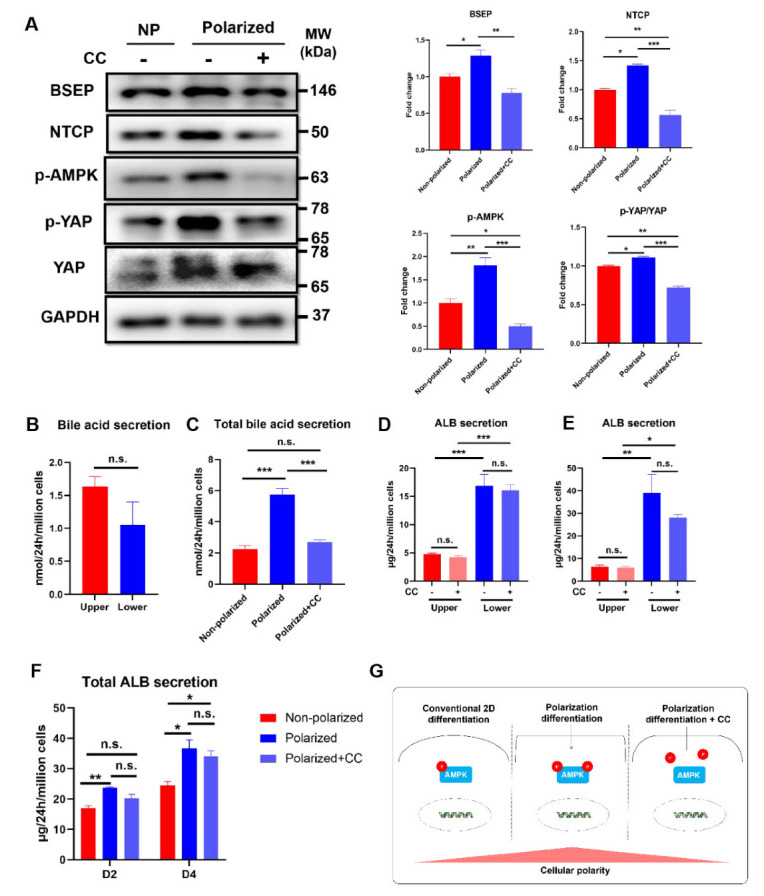
Activation of AMPK signaling pathway involved in the polarity maintenance in the polarized hEHs (**A**) Western blot analysis of cell lysates from the nonpolarized and polarized hEHs in the presence or the absence of CC for determining the protein expressions of BSEP, NTCP and phosphorylations of AMPK and YAP (left panel), quantification of these protein levels normalized to GAPDH protein (right panel). Full-length blots were presented in Appendix A. (**B**) Analysis for the directional secretion of the bile acids in the polarized hEHs treated with CC for 4 days. (**C**) Analysis for the total bile acid secretion of the non-polarized hEHs and polarized hEHs in the presence or the absence of CC for 4 days. (**D**) Analysis for directional ALB secretion of non−polarized and polarized hEHs in the presence or the absence of CC for 2 days. (**E**) Analysis for directional ALB secretion of non-polarized and polarized hEHs in the presence or the absence of CC for 4 days. (**F**) Analysis for the total ALB secretion of non-polarized and polarized hEHs in the presence or the absence of CC for 2 days and 4 days. (**G**) Schematic representation of the activation of the AMPK signaling pathway involved in the polarity maintenance in the polarized hEHs. Data represent the mean ± SD. * *p* <  0.05, ** *p*  <  0.01, *** *p*  <  0.001; n.s. means no significant.

## Data Availability

The data that support the findings of this study are available from the corresponding author upon reasonable request.

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
