# Peer review of "Hepatic Polarized Differentiation Promoted the Maturity and Liver Function of Human Embryonic Stem Cell-Derived Hepatocytes via Activating Hippo and AMPK Signaling Pathways"

_cells, 2022, doi:10.3390/cells11244117_

Round 1

Reviewer 1 Report

This work focus on the production and characterization of polarized stem cell-derived hepatocyte-like cells, exploring mechanisms involved in cell polarization. Improving the maturity of stem cell-derived hepatocyte-like cells is a current need in the development of hepatic in vitro models. Hepatocytes are polarized cells, presenting two membrane domains, the canalicular and the basolateral, which are essential for their functionality. Thus, having in vitro differentiation protocols able to produce polarized hepatocytes is extremely important to improve the physiological relevance of in vitro models for drug metabolism, toxicology and disease modeling studies. However, despite its relevance, several points must be addressed.

Major comments 

1.     Except for experiments presented in figure 1, the authors do not used controls for other experiments. I believe that if the authors want to claim that there is an improvement in cell maturity all the results should be compared to DE cells or undifferentiated cells.

2.     If I understood correctly, the generation of gradients in the transwells is the rationale for the production of polarized cells. I believe that this should be more carefully discussed and explained in the manuscript.

3.     Please describe in the Materials and Methods section relative qRT-PCR what is the control sample used in the 2-∆∆Ct.

4.     Please mind the formatting in Materials and Methods section “Establishment of the polarized hEHs”.

5.     Authors use 20% FBS in the hepatic differentiation medium (line 231). This FBS concentration is extremely high since FBS promotes proliferation and inhibits differentiation. Have the authors tried to use lower FBS concentrations?

6.     In figure 1A, it is stated that DMSO is added to the upper chamber while in the Materials and Methods and Results section it is said that the upper chamber has HDM with 10% FBS. Could the authors please clarify?

7.     In my understanding, in the Results section, the description of the experimental conditions made in lines 225-237 belongs to Materials and Methods section. Could the authors please incorporate this in the correspondent section of Materials and Methods for the sake of clarity?

8.     It is not clear what the authors want to show in figure 3C.

9.     In figure 4D, how do the authors explain the lower Hnf4a expression in polarized hEHs, as this is a typical mature hepatocyte marker?

10.  In figure 4I, authors claim that rifampicin induced CYP3A4 activity but the results for non-treated hEHs are not shown. 

11.  What is the concentration of rifampicin used for experiments in figures 4K-P? Is it toxic?

12.  In figure 6A, authors observe that multiple signaling pathways are enriched in polarized hHEs. Why did they choose to analyze the Hippo signaling pathway in particular? I believe that there is a contribution from all the pathways since hepatic differentiation is a complex process and this should be discussed.

13.  I believe it would improve the paper’s impact if the authors showed immunofluorescence results that demonstrated the polarization of membrane apical and basolateral transporters such as MRP2 and OATP-C instead of just evaluating the expression and presence of apical transporters (figure 3A).

Minor comments:

1.     Please state what is “HCM” in the abbreviations section.

2.     The names of the genes of figure 5 are difficult to read due to letter size.

Author Response

Response: We greatly appreciate the reviewer for his/her review and evaluation of our manuscript.

Major comments:

Comment 1. Except for experiments presented in figure 1, the authors do not used controls for other experiments. I believe that if the authors want to claim that there is an improvement in cell maturity all the results should be compared to DE cells or undifferentiated cells.

Response: We are grateful for the reviewer’s comment and constructive suggestion. Actually the major objective of this study is to compare the function and maturity between the non-polarized and polarized hEHs. Except for using the transwells, the differentiation process and used growth factors in this study are the same as our previous studies [1,2]. Moreover, we have well demonstrated that DE or undifferentiated hESCs can be efficiently differentiated into hEHs using our differentiation protocol. Based on the above reasons, we mainly chose the non-polarized hEHs as the control group for analysis in this study.

    Per request by reviewer, in order to more convincibly show the improvement of maturation and liver function in the polarized hEHs, we compared the differences in the relative expression of genes associated with hepatic maturation and function in DE cells, the non-polarized cells, the polarized cells, and human primary hepatocytes, and the results were shown as Figure 4A-E in the revised manuscript.

    In addition, based on the addition of the results from primary human hepatocytes, subsection of Isolation and culture of primary human hepatocytes was added in the Materials and Methods section at lines 6-20 on page 8 in the revised version.

References

[1] Duan Y., et al. Differentiation and Characterization of Metabolically Functioning Hepatocytes from Human Embryonic Stem Cells. Stem cells 2010; 28:674–686.

[2] Chen J., et al. Salvianolic Acid B Enhances Hepatic Differentiation of Human Embryonic Stem Cells through Upregulation of WNT Pathway and Inhibition of Notch Pathway. Stem Cells Dev 2018; 27:252-261.

Comment 2. If I understood correctly, the generation of gradients in the transwells is the rationale for the production of polarized cells. I believe that this should be more carefully discussed and explained in the manuscript.

Response: We really appreciate the reviewer for his/her constructive suggestion. Per suggestion by reviewer, and in order to further discuss the reasons why the hEHs acquired polarized states on transwells, the explanation was added at 14-20 lines on page 24 in the Discussion section in revised manuscript.

Comment 3. Please describe in the Materials and Methods section relative qRT-PCR what is the control sample used in the 2-∆∆Ct.

Response: We really appreciate the reviewer for his/her constructive suggestion. Per suggestion by the reviewer, the description about the relative gene expression using 2-ΔΔCt method was added in the Materials and Methods section at 14-16 lines on page 10 in revised version.

Comment 4. Please mind the formatting in Materials and Methods section “Establishment of the polarized hEHs”.

Response: We are very sorry for formatting errors in Materials and Methods section, and we corrected it at 15 line on page 6 in the revised manuscript.

Comment 5. Authors use 20% FBS in the hepatic differentiation medium (line 231). This FBS concentration is extremely high since FBS promotes proliferation and inhibits differentiation. Have the authors tried to use lower FBS concentrations?

Response: We are grateful for the reviewer’s comment and constructive suggestion. The FBS concentration used in this study are based on our previous studies [1-3]. In fact, the purpose of using 20% FBS is to increase the density of differentiated cells by promoting cell proliferation, because the increased cell density will help improve the liver function of the hEHs or adult hepatocytes. However, we did not pay our attention to the potential inhibitory effect of high FBS concentration on cell differentiation. According to constructive suggestion of the reviewer, we will reduce the concentration of FBS in our future studies, and find an appropriate FBS concentration to balance the proliferation and differentiation during the differentiation of hESCs.

References

[1] Duan Y., et al. Differentiation and Characterization of Metabolically Functioning Hepatocytes from Human Embryonic Stem Cells. Stem cells 2010; 28:674–686.

[2] Chen J., et al. Salvianolic Acid B Enhances Hepatic Differentiation of Human Embryonic Stem Cells through Upregulation of WNT Pathway and Inhibition of Notch Pathway. Stem Cells Dev 2018; 27:252-261.

[3] Chen J., et al. Enhancement of Hepatocyte Differentiation from Human Embryonic Stem Cells by Chinese Medicine Fuzhenghuayu. Sci Rep 2016;6:18841.

Comment 6. In figure 1A, it is stated that DMSO is added to the upper chamber while in the Materials and Methods and Results section it is said that the upper chamber has HDM with 10% FBS. Could the authors please clarify?

Response: We deeply apologize for this error. We had updated the Figure 1A, all the differentiated information were labeled and listed in the new one. The previous Figure 1A was replaced by the updated Figure 1A in the revised manuscript.

Comment 7. In my understanding, in the Results section, the description of the experimental conditions made in lines 225-237 belongs to Materials and Methods section. Could the authors please incorporate this in the correspondent section of Materials and Methods for the sake of clarity?

Response: We really appreciate the reviewer for his/her constructive suggestion. Per the suggestion by the reviewer, we removed the contents in lines 225-237 to Materials and Methods section, and modified the description from 13 lines on page 7 to 4 lines on page 8 in our revised manuscript.

Comment 8. It is not clear what the authors want to show in figure 3C.

Response: We are very sorry for making the confusion for the reviewer. In normal physiological condition, F-actin and BSEP are relatively concentrated on the canalicular membrane of hepatocytes in the human liver tissue, thus we wanted to define the apical membrane side of polarized cells by showing the co-localization of BSEP and F-actin in the Figure 3C.

To illustrate this purpose more clearly, the previous description “ In addition, according to the confocal images, BSEP were mainly localized on the upper (apical) membrane in the polarized hEHs, while most of BSEP were localized with the nucleus in the non-polarized hEHs (Figure 3C).” was changed to “To determine the pattern of cellular polarity, we performed co-immunostaining of hEHs with F-actin and BSEP (relatively concentrated on the canalicular membrane of hepatocytes in the human liver tissue) and analyzed their z-stack confocal images. The results of the confocal images showed that F-actin and BSEP were mainly co-localized on the upper (apical) membrane in the polarized hEHs, while most of BSEP were localized with the nucleus and F-actin were distributed in the lateral membrane of the cell in the non-polarized hEHs (Figure 3C).” from 1 line to 8 line on page 16 in the Results section in revised manuscript. In addition, we also adjusted the brightness of Figure 3C which was provided in revised manuscript.

Comment 9. In figure 4D, how do the authors explain the lower Hnf4a expression in polarized hEHs, as this is a typical mature hepatocyte marker?

Response: We are grateful for the reviewer’s comment. HNF4α acts both the nuclear receptor and the liver-associated transcription factor, and plays an important role in hepatic differentiation. We agree with the reviewer, and also regard HNF4α as a typical mature hepatocyte marker, especially when HNF4α works as the nuclear receptor. However, HNF4α gene expression was really decreased with the establishment of polarization state in this study. DeLaForest et al. found that the HNF4α gene expression level was decreased during the mature stage of hEHs [1], and the same findings were also confirmed by Zhao’s study [2], the HNF4α gene expression level of mature hEHs was also down-regulated than that of the immature hEHs. In addition, a recent study reported that HNF4α was an accepted marker of human hepatic progenitor cells [3], based on this finding, the decreased HNF4α gene expression level might indicate the increased maturity of the polarized hEHs in our present study. However, whether the decreased HNF4α gene expression level can be used to evaluate the maturity of hEHs still needs to be further verified. In order to avoid unnecessary controversy, the result of HNF4α gene expression was removed, and replaced by the gene expression level of CK19 (a hepatic progenitor marker) as Figure 4E in revised manuscript.

References

[1] DeLaForest Ann., et al. HNF4A is essential for specification of hepatic progenitors from human pluripotent stem cells. Development 2011;138(19): 4143-4153.

[2] Zhao D., et al. Promotion of the efficient metabolic maturation of human pluripotent stem cell-derived hepatocytes by correcting specification defects. Cell research 2013; 23(1): 157-161.

[3] Dao T., et al. Stem cell-derived polarized hepatocytes. Nature communications 2020; 11(1): 1-13.

Comment 10. In figure 4I, authors claim that rifampicin induced CYP3A4 activity but the results for non-treated hEHs are not shown.

Response: We are grateful for the reviewer’s comment. Per request by the reviewer, the results of the non-treated hEHs without the induction by rifampicin were added as new Figure 4I in our revised manuscript.

Comment 11. What is the concentration of rifampicin used for experiments in figures 4K-P? Is it toxic?

Response: We are grateful for the reviewer’s comment. In Figures 4K-P, the concentrations of rifampicin used for experiments were 25 μM rifampicin. This concentration of rifampicin was used to induce the response of human pluripotent stem cell-derived hepatocytes in our previous study [1], and was not toxic to cells.

References:

[1] Ma X., et al. Highly efficient differentiation of functional hepatocytes from human induced pluripotent stem cells. Stem Cells Translational Medicine 2013; 2:409-419.

Comment 12. In figure 6A, authors observe that multiple signaling pathways are enriched in polarized hHEs. Why did they choose to analyze the Hippo signaling pathway in particular? I believe that there is a contribution from all the pathways since hepatic differentiation is a complex process and this should be discussed.

Response: We really appreciate the reviewer for his/her comment and constructive suggestion. The Hippo signaling pathway is an evolutionarily conserved signaling module, and it is closely related to liver size control, regeneration, development and tumorigenesis. In particular, the many components of Hippo pathway are apically localized, which is important for their activity [1]. Borreguero-Muñoz N et al confirmed that the Hippo pathway had a physiological function as an integrator of epithelial cell polarity, tissue mechanics, and nutritional cues to control cell proliferation and tissue growth in both Drosophila and mammals [2]. In this study, our purpose was to develop an approach to generate polarized hepatocytes, and also to attempt to reveal the mechanism by which regulated the polarization in our polarized hepatocytes, thus, this is the major reason why we particularly chose Hippo signaling pathway for verification.

    The aforementioned description about the reason why we chose Hippo signaling pathway was added in the Discussion section at 9-14 lines on page 27 in revised version.

In addition, we agree with the reviewer’s opinions, the improvement of maturation and function of hEHs was associated with the activation or inactivation of different signaling pathways. Our previous study revealed that activation of Wnt pathway and inhibition of Notch pathway enhanced hepatic differentiation of hHEs [3,4]. In addition, Gao et al. demonstrated that inhibiting the MAPK/ERK signaling pathway, and subsequently attenuating the WNT signaling pathway negatively regulated hepatic differentiation of hESC-derived hepatic progenitors [3]. In agreement with these results, we found that MAPK and Wnt signaling pathways were remarkable enriched based on the RNA-seq data in this study. These results indicated that hepatic differentiation was a complex process, and the polarization differentiation culture could extensively affect the different signaling pathways during these processes.                                                     .

The previous description of “In our previous studies, we have revealed that the activation of MAPK and Wnt signaling could improve the efficiency of hepatic differentiation and function in hEHs, in agreement with these results, we found that MAPK and Wnt signaling pathways were remarkable enriched based on the RNA-seq data in this study.” was changed to “Our previous study revealed that activation of Wnt pathway and inhibition of Notch pathway enhanced hepatic differentiation of hHEs. In addition, Gao et al. demonstrated that inhibiting the MAPK/ERK signaling pathway, and subsequently attenuating the WNT signaling pathway negatively regulated hepatic differentiation of hESC-derived hepatic progenitors. In agreement with these results, we found that MAPK and Wnt signaling pathways were remarkable enriched based on the RNA-seq data in this study. These results indicated that hepatic differentiation was a complex process, and the polarization differentiation culture could extensively affect the different signaling pathways during the polarized differentiation.” at lines 20-22 on page 26 and 1-6 lines on page 27 in our revised manuscript.

References

[1] Borreguero-Muñoz N. et al. The Hippo pathway integrates PI3K–Akt signals with mechanical and polarity cues to control tissue growth. PLoS biology 2019;17(10): e3000509.

[2] Yang C. et al. Differential regulation of the Hippo pathway by adherens junctions and apical–basal cell polarity modules. PNAS 2015;112(6): 1785-1790.

[3] Gao W. et al. Ethanol negatively regulates hepatic differentiation of hESC by inhibition of the MAPK/ERK signaling pathway in vitro. PloS one 2014; 9(11): e112698.

[4] Chen J., et al. Enhancement of hepatocyte differentiation from human embryonic stem cells by Chinese medicine Fuzhenghuayu. Sci Rep 2016; 6:18841.

Comment 13. I believe it would improve the paper’s impact if the authors showed immunofluorescence results that demonstrated the polarization of membrane apical and basolateral transporters such as MRP2 and OATP-C instead of just evaluating the expression and presence of apical transporters (figure 3A).

Response: Again, we really appreciate the reviewer for his/her comment and constructive suggestion. We totally agree with reviewer's opinion. Indeed, we attempted to observe the distribution of NTCP (a typical basal membrane marker) in the polarized hEHs. Unfortunately, the basal membrane was not depicted by our confocal microscope settings due to the optical interference of the Transwell membrane. Hayashi H et al. also encountered the same technical problem when conducting research with transwells [1].

References

[1] Hayashi H. et al. Modeling human bile acid transport and synthesis in stem cell-derived hepatocytes with a patient-specific mutation. Stem cell reports 2021; 16(2): 309-323.

Minor comments:

Comment 1.  Please state what is “HCM” in the abbreviations section.

Response: We are grateful for the reviewer’s suggestion. The abbreviation of HCM stands for hepatocyte culture medium, and it was added in the abbreviations section in revised version.

Comment 2.  The names of the genes of figure 5 are difficult to read due to letter size.
Response: We are grateful for the reviewer’s comment. We updated these images by enlarging the size of the gene names in the heat map. The previous Figure 5 was replaced by updated one in revised manuscript.

Reviewer 2 Report

In this article Wang et al. describe a new method for culturing human hepatocytes derived from human embryonic stem cells. The authors claim that their method promotes polarized differentiation in hepatocytes, leading to hepatocyte maturation. Further RNA sequencing analysis indicated significant alterations in genes involved in cell adhesion, transporters, and CYP450. A KEGG pathways analysis determined that these “polarized” hepatocytes showed alterations in genes involved in MAPK, Hippo, and Wnt signaling. To strengthen this manuscript, the authors should address the following:

1.     The schematic in Figure 1A does not match the described method.

2.     I believe you are misnaming cytokines. These instead are growth factors and not cytokines.

3.     The first figure is aiming to solidify that these cells have polarity and are differentiated hepatocytes. However, I am not convinced that this figure strongly show either.  The images in this figure are difficult to see and polarity might be better visualized with beta actin or e-cadherin, while differentiation would be better assessed with staining of cytokeratin 18 & 19.

4.     Overall, you compare your polarized to non-polarized. If the intention is to recapitulate adult liver hepatocytes, you should compare your in-vitro samples with DE and adult liver. This will be more convincing for maturation analysis.

5.     Did you starve your cells before analysis? Many of these growth factors will affect your data. If you did, please write into the methods.

6.     The FITC data is a little confusing, I would remove this. I would instead use TEER readings, if you can get them.

7.     For Figure 3, MDR1 protein should be quantified, along with a basolateral marker (i.e. SCRIB).

8.     The images in Figure 3 are a little blurry and hard to see.

9.     The RNA sequencing should also be compared to endoderm, adult liver, and possibly fetal liver.

10.  I know that you state that Compound CC is an AMPK inhibitor, but you don’t describe the dose, where its from, etc in the methods. Please do this.

11.  There are inconsistent front sizes, please correct.

Author Response

Response: We greatly appreciate the reviewer for his/her review and evaluation of our manuscript.

To strengthen this manuscript, the authors should address the following:

Comment 1. The schematic in Figure 1A does not match the described method.
Response: We deeply apologize for this error. We have updated the Figure 1A, all the information about the differentiation were labeled in the updated Figure 1A which replaced the previous one in our revised manuscript.

Comment 2. I believe you are misnaming cytokines. These instead are growth factors and not cytokines.
Response: We are very sorry for making this mistake. Per request by the reviewer, the misnaming cytokines have changed in our revised manuscript.

Comment 3. The first figure is aiming to solidify that these cells have polarity and are differentiated hepatocytes. However, I am not convinced that this figure strongly show either. The images in this figure are difficult to see and polarity might be better visualized with beta actin or e-cadherin, while differentiation would be better assessed with staining of cytokeratin 18 & 19.
Response: We really appreciate the reviewer for his/her comment and constructive suggestion. The major reason why we showed the first picture was to illustrate that the developed polarization differentiation did not affect hepatocyte differentiation from hEHs, thus, we did not show the polarity of hEHs in this figure, afterwards we fully revealed that hEHs had typical polarization characteristics using the polarization differentiation approach in the subsequent figures (figures 2-6). In normal physiological condition, F-actin and BSEP were relatively concentrated on the canalicular membrane of hepatocytes in the human liver tissue, thus we defined the apical membrane side of polarized cells by showing the co-localization of BSEP and F-actin rather than using beta actin or e-cadherin in the Figure 3C.

    In addition, considering cytokeratin 18 & 19 also are expressed in other epithelia cells, we always use hepatocyte specific markers (human albumin (ALB) and alpha-1 antitrypsin (a1-AT)) to monitor and assess hepatocyte differentiation.

Comment 4. Overall, you compare your polarized to non-polarized. If the intention is to recapitulate adult liver hepatocytes, you should compare your in-vitro samples with DE and adult liver. This will be more convincing for maturation analysis.
Response: We are grateful for the reviewer’s comment and constructive suggestion. The major objective of this study is to compare the function and maturity between the non-polarized and polarized human embryonic stem cells-derived hepatocytes (hEHs). Per request by the reviewer, in order to more convincibly show the improvement of the maturation and liver function in the polarized hEHs, we compared the differences in the relative expression of genes associated with hepatic maturation and function in DE cells, the non-polarized cells, the polarized cells, and primary human hepatocytes, and the results were shown as Figure 4A-E in revised manuscript.

    In addition, based on the addition of the results from primary human hepatocytes, subsection of Isolation and culture of primary human hepatocytes was added in the Materials and Methods section at lines 6-20 on page 8 in the revised version.

Comment 5.  Did you starve your cells before analysis? Many of these growth factors will affect your data. If you did, please write into the methods.

Response: We are grateful for the reviewer’s comment. We did not starve the cells before analysis, normally the cells were harvested 48 hours after the medium were changed, considering the half-life period and consumption by the cells, the amount of those growth factors was dramatically reduced at the time of harvesting cells. In addition, the amount of those growth factors inside the cells would be also decreased with the reduction of growth factors in the culture medium. This is very interesting point, we will starve the cells before analysis and make the comparison in our future study. 

Comment 6. The FITC data is a little confusing, I would remove this. I would instead use TEER readings, if you can get them.

Response: We are very sorry for making the confusion for the reviewer. In order to verify directional secretion by the polarized hEHs, we assessed the paracellular permeability of the polarized hEHs in the transwells using the 4kDa FITC-labeled dextran. As shown in Figure 2F, whatever FITC-labeled dextran was placed in the upper chambers or the lower chambers, FITC-fluorescence signals were not detected in the opposite chambers of transwells in the presence of the polarized hEHs, whereas either the upper or the lower chambers had equal fluorescence signals in the absence of the polarized hEHs, indicating such directional secretion was not caused by the paracellular permeability. As a result, this phenomenon demonstrated that the directional secretion of hepatic cargos was due to the active transportation of the polarized hEHs rather than their paracellular diffusion.

We agree with the reviewer, transepithelial electrical resistance (TEER) readings are an important index for evaluating the formation of single cell layer, however, we do not have the Millicell-ERS device, thus, technically we are not able to use   TEER reading in this study.

Comment 7. For Figure 3, MDR1 protein should be quantified, along with a basolateral marker (i.e. SCRIB).

Response: We really appreciate the reviewer for his/her comment and constructive suggestion. Per request by the reviewer, organic aniontransporting polypeptide 1 (OATP1), a hepatic basolateral marker, was quantified and presented in Figure 3B, and the description on the results was added at XX on page 15 in Results section in our revised manuscript. However, as to basolateral marker SCRIB, currently we do not have primers and antibody to evaluate it, and used OATP1 instead.  

Comment 8. The images in Figure 3 are a little blurry and hard to see.

Response: We are very sorry for making the confusion for the reviewer. In human liver tissue, F-actin and BSEP are relatively concentrated on the canalicular membrane of hepatocytes, thus we wanted to define the apical membrane side of polarized cells by showing the co-localization of BSEP and F-actin. Thus, we adjusted the brightness of the images of Figure 3C in the revised manuscript.

Comment 9.  The RNA sequencing should also be compared to endoderm, adult liver, and possibly fetal liver.

Response: We are grateful for the reviewer’s comment and constructive suggestion. The major objective of this study is to compare the function and maturity between the non-polarized and polarized hEHs. Based on this purpose, we chose the non-polarized hEHs as the control group for RNA sequencing in this study. In addition, in order to more convincibly show the improvement of maturation and hepatic function in the polarized hEHs, we compared the differences in the relative expression of genes associated with hepatic maturation and function in DE cells, the non-polarized cells, the polarized cells, and human primary hepatocytes, and the results were presented in Figure 4A-E in the revised manuscript.

Comment 10.  I know that you state that Compound CC is an AMPK inhibitor, but you don’t describe the dose, where its from, etc in the methods. Please do this.

Response: We really appreciate the reviewer for his/her suggestion. Per suggestion by the reviewer, the concentration and the origination of Compound CC were added at 2-4 lines on page 8 in Materials and Methods section of the revised manuscript.

Comment 11.  There are inconsistent front sizes, please correct

Response: We deeply apologize for formatting errors, and we checked the front sizes of the entire manuscript. In addition, sometimes the PDF conversion during the manuscript submission may cause these errors, and we will re-check it at the submission.